# Pilot Study of the Applicability, Usability, and Accuracy of the Nutricate© Online Application, a New Dietary Intake Assessment Tool for Managing Infant Cow’s Milk Allergy

**DOI:** 10.3390/nu15041045

**Published:** 2023-02-20

**Authors:** Pauline Azzano, Line Samier, Alain Lachaux, Florence Villard Truc, Laurent Béghin

**Affiliations:** 1Department of Pediatric Hepatogastroenterology and Nutrition, Hôpital Femme Mère Enfant, Hospices Civils de Lyon, INSERM U1111, Lyon 1 University, Bron, F-69500 Lyon, France; 2Nutricia Danone, 17 Rue des 2 Gares, Rueil-Malmaison, F-92500 Paris, France; 3Primary Care Pediatrician Office, 28 Avenue Rockfeller, F-69008 Lyon, France; 4INSERM U1286—INFINITE—Institute for Translational Research in Inflammation, and CIC1403—Clinical Investigation Center, CHU Lille, Lille University, F-59000 Lille, France

**Keywords:** cow’s milk allergy, dietary intake, dietary online application

## Abstract

Background/Objectives: The mainstay treatment of cow’s milk allergy (CMA) is to remove cow’s milk proteins from children’s dietary intake. In this context, dietary intake of children with CMA should be particularly checked and monitored. The objective of this study was to assess the applicability, usability, and accuracy of a new dietary intake (DI) assessment online tool (Nutricate© online application) for managing CMA in children. Subjects/Methods: This study used a pre-existing database of DI from the Nutricate© online application. DIs from 30 CMA children were used to compare micro/macronutrients (energy, protein, calcium, and iron intakes) calculated by Nutricate© and NutriLog© as the reference method. Comparisons were performed using the Pearson correlation analysis and the Bland–Altman plot. The Nutricate© tool usability was assessed via a System Usability Scale questionnaire (SUSq). Results: Correlation coefficient between the levels of micro/macronutrients obtained by Nutrilog© and Nutricate© software were highly significant (*p* = 0.0001) and were well-correlated (R coefficient > 0.6), indicating a very good concordance between the two methods. This observation was reinforced by the Bland–Altman plot, indicating the absence of proportional or fixed bias for energy, protein, calcium, and iron intakes. The mean SUSq score obtained was 81 ± 14, which is considered to be an excellent score. Conclusions: Nutricate© online application is a reliable method to assess micro/macronutrient (energy, protein, calcium, and iron intakes) intake in CMA children. Applicability and usability of this new dietary intake assessment online tool is excellent.

## 1. Introduction

Cow’s milk allergy is the most common type of food allergy in early childhood. In industrialized countries, this allergy affects 2–7% of formula-fed infants [1] and 3.4% in France [2]. Cow’s milk allergy is an immune-mediated reaction to proteins from cow’s milk [3]. The mainstay treatment consists of removing cow’s milk proteins from infant dietary intake. The exclusion of the cow’s milk protein is needed to prevent the occurrence of complex symptoms, such as severe reactions, faltering growth, atopic comorbidities, and non-specific symptoms, such as vomiting, diarrhea, colic, rash/urticaria, and gastrointestinal bleeding [4]. Whilst breastmilk remains the ideal nutrient source in infants with cow’s milk allergy, infants not exclusively breastfed require a hypoallergenic formula (HAF), which includes extensively hydrolyzed formulas (eHF), hydrolyzed rice formulas (HRF), soy formulas (not available in the French market), or amino acid formulas (AAF) [3,5,6]. In this context, cohort studies have shown malnutrition to be commonly observed in infants with cow’s milk allergy during the exclusion diet, as well as in those newly diagnosed with cow’s milk allergy [7]. Indeed, cow’s milk allergy may lead to a decrease in some micro/macronutrient intakes that are crucial for infants’ growth and development [8,9], in part due to a lack of parents’ knowledge regarding the dietary recommendations and management, hence leading to nutritional deficiencies and imbalance [6]. Based on this gap, there is a need to use a dietary survey on infants with cow’s milk allergy as a key element for cow’s milk allergy management. Clearly, a dietician’s input is highly recommended to maintain the optimal growth of infant by providing nutritional advice and counseling and by guiding the choice of a milk substitute [10]. 

Cow’s milk allergy is mainly managed by primary care pediatricians (PCP), who are not necessarily specialized in allergology or familiar with nutritional management [11]. Most often, patients with cow’s milk allergy require specific dietary management; unfortunately, PCPs often have no access to a dietician or to a specific dietary tool for cow’s milk allergy management. Consequently, the patient is most often referred to a hospital dietician or an allergy clinic [10]. Cow’s milk allergy management has an economic impact that generates an increase in health costs due to the duplication in medical consultation between the PCP, the hospital [12], and the other health services, such as specialized nutrition [13]. Several studies have highlighted the clinical [14], population-level [15], and economic burden of cow’s milk allergy [11]. Indeed, medical consultations in a hospital setting are time-consuming for both healthcare professionals and parents. In addition, dietary surveys require data collection, data management, and interpretation of a specific dietary software by a well-trained dietician [16]. Moreover, the PCP has to wait for all these steps before receiving the data to guide him on treatment options [17]. Clearly, all these constraints make it necessary to use a practical and efficient tool for cow’s milk allergy dietary management. Therefore, accessing the patient’s nutritional intake data during the consultation may help to identify and correct possible deficiencies and orient the diagnosis. For example, in case of weight stagnation, it will allow the identifying of a deficiency in energy intake or an allergy to cow’s milk proteins coupled with an allergy to protein hydrolysates, requiring the use of an amino acid formula.

Use of computer-assisted dietary collection tools has already been recognized [18]. Recently, Nutricia© designed the Nutricate© tool, a new online application for dietary intake assessment specifically designed for cow’s milk allergy dietary management using a web environment (www.aplv.fr (accessed on 15 June 2022)). Unlike other pathologies, cow’s milk allergy dietary management is mainly focused on specific nutritional needs, such as energetic needs; macronutrients, such as protein intake; micronutrient needs [8], such as calcium to prevent bone mineral density; [19] and iron intake to prevent iron deficiency [20]. Based on this, the innovation of the Nutricate© tool offers some advantages, such as shortened food and beverages categories, an analysis report, a user-friendly interface, and secure data storage and retrieval.

The main objective of this study was to assess the applicability, usability, and accuracy of the Nutricate© web application, a new dietary intake assessment online application for cow’s milk allergy management.

## 2. Materials and Methods

### 2.1. Sample

The Nutricate© web application was launched online in January 2020. Since the launch, the dietary intake data of 226 children with cow’s milk allergy were entered by either the PCPs or the parents. 

For the purpose of this study, between November 2020 and May 2022, the dietary intake data entered by volunteer parents of 30 children with cow’s milk allergy were used. The time lapse between the age at diagnosis and this present analysis was, on average, 40.4 ± 24.5 months. All children and their parents participating in this study were screened and included by the same PCP (FV).

Given this research was not interventional or meant to improve biological or medical conditions, but rather relied on using a pre-existing database, the present study was therefore not considered interventional research, according to French regulatory requirement (“Jardé” French law). In this context, this study did not require any approval from an ethics committee [21]. This study was, however, declared and approved by the French commission in charge of the safeguard of personal data protection (Commission Nationale de l’Informatique et des Libertés) under the reference CNIL Nutricate_Kids HCL 21_5238/2020.

Dataset related to anthropometrics and clinical phenotypes (other allergies and symptoms) of the 30 participants with cow’s milk allergy were anonymously collected by one PCP (FV) on a secure Excel file. An audit of the complete dataset was performed before statistical analysis.

### 2.2. Participants’ Anthropometric Characteristics

Body weight was measured by using an electronic scale to the nearest 0.1 kg while ensuring the participant was wearing light clothes without shoes. Height was measured, without shoes, by using a standard physician’s scale to the nearest 0.1 cm. Body mass index (BMI) was calculated by taking the participant’s weight in kilograms divided by the square of the height in meters (kg/m^2^). Z-score for weight-for-age, height-for-age, and BMI-for-age were classified according to the WHO’s classification [22] by using an anthropometric Z-score calculator R package [23]. Anthropometric characteristics and clinical phenotypic data related to the 30 children with cow’s milk allergy are presented on Table 1. Participants were well-balanced according to gender, and Z-scores were close to the normality. Clinical phenotypes were in line with what was expected of patients with cow’s milk allergy.

### 2.3. Nutricate© Software

Nutricate© is a user-friendly web application developed by Nutricia (LS) by adhering to guidelines related to the data collection of dietary information, with the goal of estimating the levels of some nutrient intakes [24]. The use of subject-based computerized food data intake was previously used in the HELENA study [25]. The Nutricate© web application collects dietary intake data (infant formula, nutritional supplements, and foods and beverages) based on the 24-hour dietary recall (24HDR) method. Parents were asked to report the daily average of food intake over a one-week period prior to the collection date. The 24HDR collection frame was presented based on a standardized meal sequence of breakfast, lunch, and dinner (Figure 1a). This 24HDR data collection frame was previously used by Thompson et al. [26]. Each meal sequence presented pictures of increasing portion size (Figure 1b) using the same process developed by Hercberg et al. [27] containing pictures of several foods and beverages [27,28]. After 24HDR completion and validation by parents, the Nutricate© web application automatically generated a password. This password allowed access to an individual dietary report with specific macro/micronutrient intake values (Table 2). The quantities of each food and beverage category reported in the Nutricate© software was automatically linked to the French food composition database (CIQUAL/ANSES food table; https://ciqual.anses.fr/ accessed on the 10 July 2020). In order to present a short list of foods, some foods were merged into one category (presented in Appendix A with their correspondence in the French food composition database). For instance, all animal-protein-rich foods (i.e., white/red meat, fish, ham, and egg) were merged into one category. The repartition of all animal-protein-rich foods was computed according to one category, similar to the repartition used in the INCA study of French infants (0–6 years) [29]. In this context, Appendix A presents correspondence between some created, merged food categories and their macronutrient contents. Appendix A presents present data for animal-protein-rich foods, starch, vegetables, and fruits.

### 2.4. Nutricate© Web Application: Usability Metrics

Dietary intake data from a subset of infants (n = 11) were randomly selected to measure the user’s satisfaction and the tool usability through the System Usability Scale questionnaire (SUSq) from Brooke [30]. The System Usability Scale questionnaire is a standard, non-proprietary questionnaire designed to be both simple and quick. The questionnaire consists of 10 questions to evaluate the complexity, usability, and user-friendliness of the software. The System Usability Scale questionnaire uses the Likert scale. For each of the 10 questions asked, the user is invited to choose between 5 possible answers ranging from “Totally disagree” to “Totally agree”, with 1 referring to poor usability and 5 referring to a high usability. The computation of the System Usability Scale questionnaire score was performed according to Bangor et al. [31]. The System Usability Scale questionnaire was initially developed by Brooke [30] as a “quick” survey scale that would allow practitioners to quickly and easily assess the usability of a product or service with a human–computer interface. The System Usability Scale questionnaire measures how a product/service/tool can be used by the users to achieve specific goals with effectiveness, efficiency, and satisfaction in the context of a given use. The System Usability Scale questionnaire has been in use for approximately 30 years and is a reliable tool. The System Usability Scale questionnaire has become an industry standard, with references in over 1300 articles and publications. The System Usability Scale questionnaire is defined by the International Organization for Standardization (ISO) as “the extent to which a product can be used by certain users to achieve specific goals with effectiveness, efficiency and satisfaction in a context of specific use” (ISO/IEC 9241-11 of 1998) and as “usability refers to the ability of a software being understood, learned, used and being attractive to the user, under specific conditions of use” (ISO/IEC 9126-1 of 2001).

At the beginning, the System Usability Scale questionnaire was mainly used as an example for testing interactive voice response systems (IVRs). As the use of internet-based tools became very widely used over the past years, the System Usability Scale questionnaire was then used for novel hardware platforms and, progressively, for internet platforms. The main advantage of the System Usability Scale questionnaire is that it provides a single score on a scale that is easily understood by a wide range of people (projects managers, software engineers, and informatic programmers/developers). In this context, the use of SUSq is very useful for facilitating the software development process. The main objective is to enhance the efficiency of the human–computer interface and the learning process of the practioner to use this new computer-based system/tool. The Nutricate© web application was initially designed by a dietician (LS) and internal project managers, software engineers, and informatic programmers/developers. In this context, the use of the System Usability Scale questionnaire is very relevant to test the usability, agility, and comfort of the new Nutricate© web application.

There are many benefits to using the System Usability Scale questionnaire because this questionnaire is very easy to administer to participants, can be used on small sample sizes with reliable results (10 parents in this study), and its validity allows for the differentiation between usable and unusable systems. Moreover, the questions are short, highly comprehensive, easily customizable, and easily administered via simple survey internet tools.

In addition to this questionnaire, the education levels (EL) of parents participating in this study were assessed by using the International Standard Classification of Education (ISCED) (http://www.uis.unesco.org/Library/Documents/isced97-en.pdf, accessed on 27 August 2022). EL was initially reported to be four groups: primary education (ISCED level 0 or 1; score = 1); lower secondary education (ISCED level 2; score = 2); higher secondary education (ISCED level 3 or 4; score = 3); and tertiary education (ISCED level 5 or 6; score = 4). For the purpose of the present study, EL was simplified by merging the two lower levels into one group (i.e., ‘primary education and lower secondary education’) and hence obtaining three groups: low EL; medium EL; and high EL.

### 2.5. 24-Hour Recall Questionnaire as a Reference

Dietary intake data for each of the 30 study participants with cow’s milk allergy, collected using the Nutricate© web application, were extracted as a list of each individual’s dietary report (example Figure 2). Each dietary report was then entered into the Nutrilog© software (version 2.31; https://nutrilog.com; accessed on 15 December 2022). Nutrilog© is a professional nutrition web application that uses the CIQUAL/ANSES French Food Composition Database (Agence Nationale de SEcurité Sanitaire de l’alimentation, de l’Environnement et du travail/French National Agency of Health, Safety of Nutrition, Environment, and Employment; https://ciqual.anses.fr; accessed on 15 December 2022). Nutrilog© was used as the reference method in this study [32]. It provided information about the individual food profile, and the food composition table was integrated into the software database containing the CIQUAL food table, France 2012, and USD SR24; USA 201 was provided by the supplier. The Nutrilog© was used to compare the micro/macronutrient outputs.

### 2.6. Statistical Analysis

As it was a pilot study, no formal sample size calculation was performed. The sample size was arbitrarily set, with thirty participants considered satisfactory enough to detect differences between the two methods (Nutrilog© and Nutricate©). Similarly, data from eleven children were considered to compute usability metrics.

Continuous variables were expressed as means (standard deviation, SD), and categorical variables were expressed as numbers (percentage). The data normality was checked using the Kolmogorov–Smirnov test and through the visual inspection of the histograms. The concordance of total energy and nutrient estimates between both collection modes (Nutrilog© and Nutricate©) were tested by Spearman correlation statistics. Bland–Altman plots were used to reinforce statistics and to evaluate bias between both methods [33]. All analyses were computed using SAS software (version 9.4, SAS Institute Inc), and *p*-values less than 0.05 were considered statistically significant.

## 3. Results

The dietary intakes of 30 children with cow’s milk allergy were recorded by their parents by using the online Nutricate© web application. After data extraction (data from Table 2), all dietary intake data were entered a second time using Nutrilog© as a reference. The mean values for the main beverages and food categories are presented in Table 3; Table 4. The Pearson correlation coefficient between Nutrilog© and Nutricate© were highly significant (*p* = 0.0001), and they were well-correlated (R coefficient > 0.6), indicating a very good concordance between the two methods (Table 5). This observation was reinforced by the Bland–Altman plot, indicating the absence of proportional or fixed bias for energy, protein, calcium, and iron intakes. The relative mean differences between Nutricate© and Nutrilog© were all <10%, with −1.7% for energy intake, +9.5% for protein intake, +9.6% for calcium intake, and −2.4% for iron intake, indicating a very good agreement between the two methods.

Bland–Altman plot data displaying the agreement between the two methods for energy, protein, calcium, and iron intakes are presented in Figure 2a–d. No proportional or fixed biases were observed for the energy, protein, calcium, and iron intakes.

Table 6 shows parents’ satisfaction parameters on the use of the Nutricate© online application, with 66.7% of users not considering data entry as time consuming and 66% reporting that the assessment of food/beverage quantity was very easy. Finally, the presence of the portion size was found to be very useful by 66% of the users. The mean duration for the completion of dietary intake was 21.8 ± 33.6 min. The mean System Usability Scale questionnaire score obtained was 81 ± 14, considered an excellent score.

Table 7 shows micro/macronutrients data (energy, protein, calcium, and iron intakes) from the Nutricate© individual dietary counseling report.

## 4. Discussion

Due to the pathology, the diversity of food group intake was less important in children with cow’s milk allergy than in children without cow’s milk allergy [34]. It was, therefore, highly relevant to propose a solution to reduce the time and process required for collecting dietary intake data for cow’s milk allergy management. The use of internet-based tools has become very widely used over the past years. In this context of digital revolution, Nutricia© developed the Nutricate© online application with a shortened list of food groups. This tool relieved the parental burden when collecting dietary intake data. The use of electronic data collection in children has previously been used; it was validated in 2002 by Baranowki et al. [35] and again in 2003 by Fletcher et al. [17]. The gold standard for dietary intake data collection consists of weighing the food and transcription into the food record [36]. The use of subject-based computerized food data intake was used more recently in the HELENA study. Moreover, it was shown that electronic records are more complete than paper-based records [37]. This study shows that Nutricate© is an easy tool to use and an accurate online application for managing children with cow’s milk allergy.

Subjects analyzed in this study were from the typical children with cow’s milk allergy population, with their anthropometric measurements (weight, height, etc.) being similar to those seen in most previously published studies [34,38] and with the food groups being similarly well-balanced, as seen previously [39]. Moreover, the levels of macronutrient intake (% carbohydrates, % proteins, and % lipids) computed from Nutrilog© (Appendix A) were similar to those observed in other cow’s milk allergy studies [34,38,40]. Furthermore, clinical phenotype characteristics of children with cow’s milk allergy from this study were similar to those of typical children with cow’s milk allergy [2]. Data related to micro/macronutrient intakes (energy, protein, calcium, and iron intakes) from the Nutricate© individual dietary counseling report (Table 7) were above 80%, showing a good adherence to cow’s milk allergy dietary recommendations. The ratio of proteins was high (265.3%), as previously observed in a previous study [41].

Correlation analysis showed a very good concordance of the Nutricate© application with Nutrilog©, the reference method, hence validating the use of this web application to manage cow’s milk allergy in children. The use of Pearson correlation analysis is the most common statistical method used to validate dietary tool measures [42]. The mean that the System Usability Scale questionnaire obtained was 81 ± 14, a score considered to be excellent [43]; this means that the Nutricate© web application is considered appropriate for the targeted population. The completion of dietary intake was considered to be fast and easy by the parents participating in this study; it was, however, interesting to note their very high level of education (data not shown), which could explain their positive feedback regarding the tool usability. This being said, the small sample size of this dataset, particularly for the System Usability Scale questionnaire score, is considered a limitation in this study. However, the strengths of this study lie on the use of solid statistical methods and a validated System Usability Scale questionnaire tool.

## 5. Conclusions

Nutricate© is a validated web application used to collect dietary intake data from children with cow’s milk allergy. Nutricate© online dietary intake collection is automatically linked to the food data composition database, allowing for a dietary report to be immediately generated. Additionally, the tool is very user-friendly; it can be used at any time and any place, which is convenient for the parents of children with cow’s milk allergy. This has the added benefits of relieving the burden of healthcare professionals by removing the need for interviewers and of data processors by removing the need to analyze dietary intake, hence saving both time and money. Most importantly, this tool shows the potential to improve the adherence of healthcare professionals to the cow’s milk allergy healthcare system. As shown in Figure 3, the Nutricate© web application has additional benefits by offering some nutritional recommendations for children with cow’s milk allergy; this tool should be tested in futures studies. Moreover, this application may be used for other pathologies.

## Figures and Tables

**Figure 1 nutrients-15-01045-f001:**
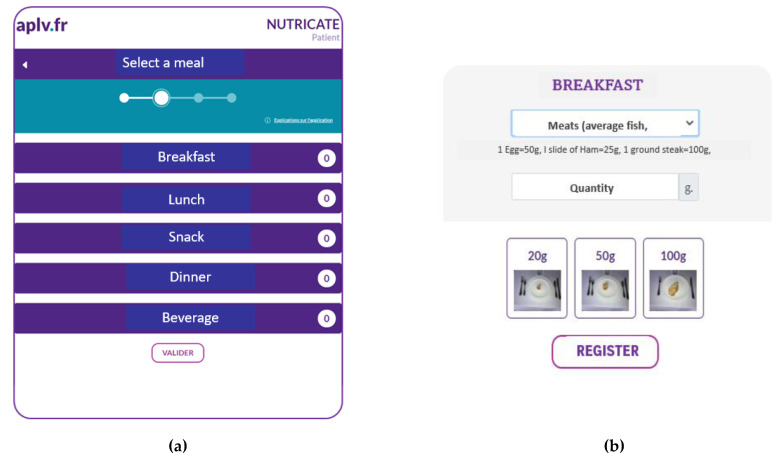
(**a**) Screenshot of the Nutricate© homepage (translated version). (**b**) Screenshot of Nutricate© food pictures with increasing portion sizes (translated version).

**Figure 2 nutrients-15-01045-f002:**
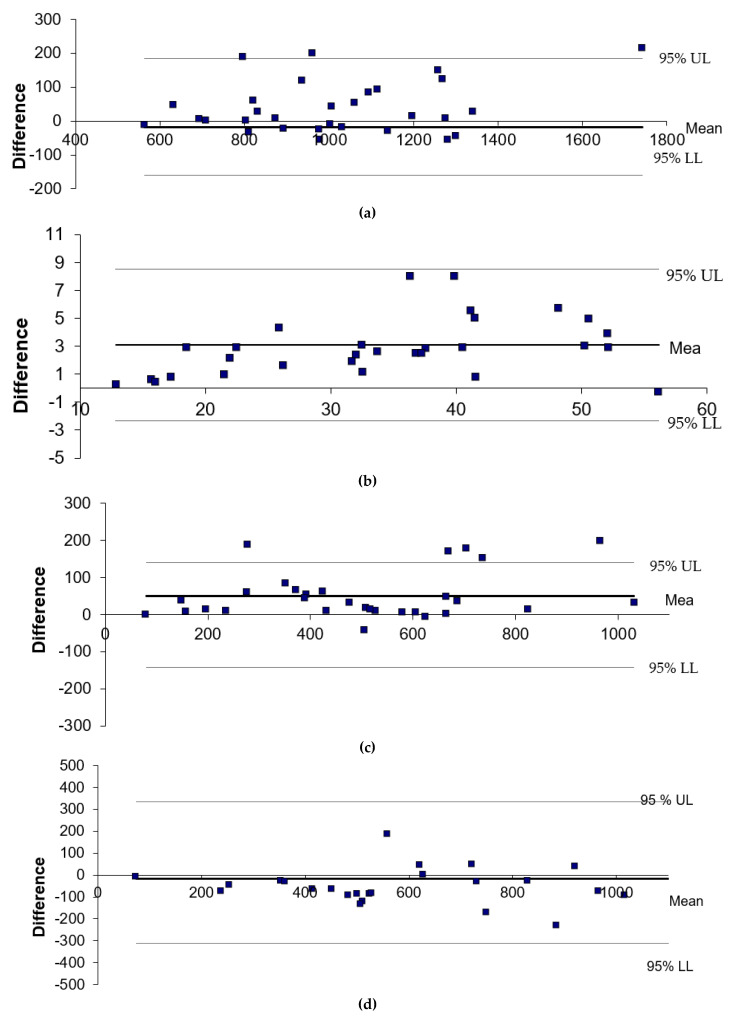
(**a**) Bland–Altman plot for energy intake between Nutricate© and Nutrilog© (kcal/d). (**b**) Bland–Altman plot for protein intake between Nutricate© and Nutrilog© (g). (**c**) Bland–Altman plot for calcium intake between Nutricate© and Nutrilog© (mg/d). (**d**) Bland–Altman plot for iron intake between Nutricate© and Nutrilog© (µg/d). The differences between the two methods were calculated as follows: Nutrilog©–Nutricate©. The 95% upper limit (UL) and lower limit (LL) of agreement (SD 1.96) are depicted as a long, dashed line. The full line indicates the mean difference and zero.

**Figure 3 nutrients-15-01045-f003:**
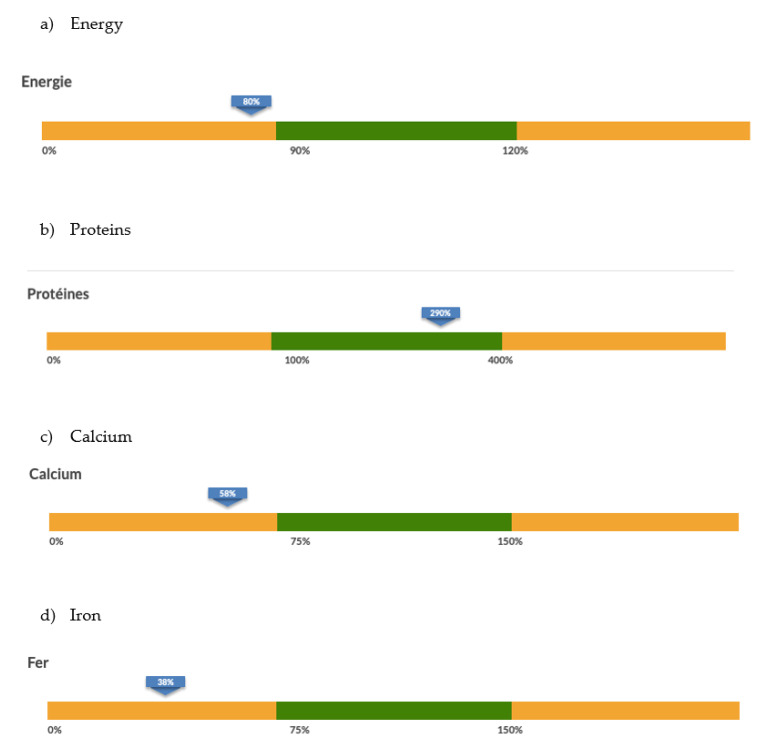
Screenshot of the Nutricate© individual dietary counseling report.

**Table 1 nutrients-15-01045-t001:** Clinical phenotypic characteristics of children with cow’s milk allergy (n = 30).

Variable	Mean ± SD
**Anthropometrics**	
Gender M/F	13/12
Age (years)	6.6 ± 1.6
Weight (kg)	23.8 ± 8.6
Height (cm)	122.2 ± 10.8
Z-score weight-for-age	−0.40 ± 1.00
Z-score height-for-age	−0.71 ± 1.43
Z-score BMI-for-age	−0.05 ± 1.30
**Other allergies and symptoms**	**n**
Other food allergies *	5/4/1/1
Asthma symptoms	9
Atopic dermatitis	7

BMI: Body mass index. * Meat/egg/rice/nut.

**Table 2 nutrients-15-01045-t002:** Example of a Nutricate© individual dietary report (translated version).

	Intake Calculation				
Results					
Food & Beverages	Quantity (G or mL)	Energy (kcal)	Proteins (g)	Calcium (mg)	Iron Absorbed (mg)
Bread (mean)	20 g	55	2	6	0.0080
Meats (mean with fish, egg, ham)	20 g	30	5	3	0.0660
Pasta, rice, semolina (cooked)	70 g	90	3	11	0.0112
oil	3 mL	27	0	0	0.0000
Vegetables raw/cooked	50 g	16	1	10	0.0050
Desert with soy, flavored, enriched with ca	90 g	86	3	108	0.0347
Biscuts (dry sweet)	2 g	9	0	1	0.0011
Bread (mean)	50 g	138	5	16	0.0200
Raw fruit, compote, and fruit puree	100 g	62	1	14	0.0060
Desert with soy, flavored, enriched with ca	90 g	86	3	108	0.0347
Meats (mean with fish, egg, ham)	20 g	30	5	3	0.0660
Margarine (non-light, unsalted)	5 g	32	0	0	0.0018
Potatoes	80 g	64	1	5	0.0080
Sweet drinks, fruit juices	90 mL	33	0	5	0.0036
**Total**		**758**	**29**	**290**	**0.2661**

**Table 3 nutrients-15-01045-t003:** Means of beverage intakes per day, according to the categories (n = 30).

Beverage Categories	Mean ± SD
	(g/Day)
Hydrolyzed formula	347.17 ± 263.35
Vegetable beverage *	200.36 ± 128.88
Fruit juices	98.89 ± 70.79
Water **	256.67 ± 333.82

* Vegetable beverage: Soja-based and other plant-based substitutes ** Water rich in calcium.

**Table 4 nutrients-15-01045-t004:** Means of solid food intake per day, according to the food categories (n = 30).

Food Categories	Mean ± SD
	(g/Day)
Cereals *	130.71 ± 89.36
Bread	40.88 ± 33.08
Breakfast cereals	25.00 ± 12.15
Infant Cereal supplements	12.00 ± 8.64
Biscuits	28.67 ± 21.95
Potatoes	72.08 ± 28.56
Vegetables	128.75 ± 89.80
Meat	65.08 ± 36.25
Oils	8.53 ± 6.28
Fruits	223.17 ± 86.05
Jam/Free sugar	12.43 ± 5.29
Black chocolate	14.14 ± 7.54

* Cereals: rice, pasta, semolina.

**Table 5 nutrients-15-01045-t005:** Pearson correlation coefficient between Nutrilog© and Nutricate© (n = 30).

Categories	R Coefficient	*p*
Energy Proteins Calcium Iron	0.631 0.978 0.957 0.914	**0.0001** **0.0001** **0.0001** **0.0001**

**Table 6 nutrients-15-01045-t006:** Nutricate© users’ satisfaction response.

Usability Item			
**Time duration for data entry**	**Not time-consuming (TC)** 66.7%	**Moderately TC** 25%	**Too TC** 8.3%
**Assessment of quantity for data entry**	**Very easy** 66.7%	**Moderately easy** 25%	**Not easy** 8.3%
**Usefulness of portion size for data entry**	**Very useful** 83.3%	**Moderately useful** 25%	**Not useful** 0%

TC: Time-consuming.

**Table 7 nutrients-15-01045-t007:** Means of some macronutrient and micronutrient intakes and their percentages, according to French recommended nutritional intake calculated by the Nutricate© software (n = 30).

Categories	Intake	Percentage (%)
Energy Proteins Calcium Iron	1038.8 ± 269.0 kcal/d 35.6 ± 12.8 g/d 527.1 ± 243.3 mg/d 709.1 ± 395.7 µg/d	88.5 ± 26.2 264.3 ± 76.8 87.4 ± 41.0 91.5 ± 53.9

## Data Availability

Not applicable.

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
