# Peer review of "Pilot Study of the Applicability, Usability, and Accuracy of the Nutricate© Online Application, a New Dietary Intake Assessment Tool for Managing Infant Cow’s Milk Allergy"

_nutrients, 2023, doi:10.3390/nu15041045_

Round 1

Reviewer 1 Report

This paper describes a pilot study that focus on the applicability, usability and accuracy of the 2 Nutricate© online application. As Authors report the use of such tools to evaluate the foods intake of children affected by CMA is not a novelty.  What makes the authors' work appreciable is the evaluation of the ease of use of the apps. 

The tip i feel to give is to review english language. For example at line 318 "Population used in this study was a typical CMA children population". It sounds redundant.  

Author Response

Reviewer #1

This paper describes a pilot study that focus on the applicability, usability and accuracy of the 2 Nutricate© online application. As Authors report the use of such tools to evaluate the foods intake of children affected by CMA is not a novelty.  What makes the authors' work appreciable is the evaluation of the ease of use of the apps. 

The tip i feel to give is to review english language. For example at line 318 "Population used in this study was a typical CMA children population". It sounds redundant.  

Many thanks for these commentaries, the English language has been reviewed by a native English speaker. The sentence on line 318 has been corrected.

Please note other modifications highlighted in yellow have been made along the manuscript to facilitate the review.

Reviewer #2

I read with interest this manuscript aiming at determining the applicability, usability and accuracy of Nutricate, an online application assessing dietary intake in cow’s milk allergic children. The article deals with an important topic given the young population under study and the effects that elimination diets can have on it. 

I suggest modifying the title simply by deleting the term “online” form one of its appearances to avoid repetitions.

Many thanks for this commentary, the title has been corrected.

The manuscript is somehow understandable although not very well written, especially in the discussion.

The English language has been reviewed by a native English speaker. Please note other modifications highlighted in yellow have been made along the manuscript to facilitate the review.

Length of the introduction is good respect to the length of the manuscript. It reviews the background and states the objectives adequately.

Many thanks for this commentary

In the discussion the results are often described as consistent and similar to those of other studies. As a clinician, I would have liked the data obtained on the intake of the various micro and macronutrients to be commented more and not only by comparing them with those obtained with other tools.

A specific paragraph has been added to address this commentary

The conclusions respond to the aim of the study.

The references are fairly recent apart from 4 papers probably kept for their unicity.

Specific remarks

Line 9 of the introduction, please add soy formulas as other possible substitutes.

Many thanks for this commentary, “soy formulas” has been added

Line 67 I wasn’t able to get the meaning of the sentence: “All CMA diagnosis were realized at least for 4 months.”

This sentence has been rephrased.

I encountered several syntax and typo errors, such as The words micro/macronutriments should be substituted by micro/macronutrients.“were closed to …” to be replaced by “were close to …”In table 1 allergia instead of allergies Supp Table 1 Jam instead of Ham Usefull instead of useful “Water rich calcium” to be replaced by Calcium rich water and many others

The English language has been reviewed by a native English speaker.

Reviewer 2 Report

I read with interest this manuscript aiming at determining the applicability, usability and accuracy of Nutricate, an online application assessing dietary intake in cow’s milk allergic children. The article deals with an important topic given the young population under study and the effects that elimination diets can have on it. 

I suggest modifying the title simply by deleting the term “online” form one of its appearances to avoid repetitions.

The manuscript is somehow understandable although not very well written, especially in the discussion.

Length of the introduction is good respect to the length of the manuscript. It reviews the background and states the objectives adequately.

In the discussion the results are often described as consistent and similar to those of other studies. As a clinician, I would have liked the data obtained on the intake of the various micro and macronutrients to be commented more and not only by comparing them with those obtained with other tools.

The conclusions respond to the aim of the study.

The references are fairly recent apart from 4 papers probably kept for their unicity.

Specific remarks

Line 9 of the introduction, please add soy formulas as other possible substitutes.

Line 67 I wasn’t able to get the meaning of the sentence: “All CMA diagnosis were realized at least for 4 months.”

I encountered several syntax and typo errors, such as

The words micro/macronutriments should be substituted by micro/macronutrients.

“were closed to …” to be replaced by “were close to …”

In table 1 allergia instead of allergies

Supp Table 1 Jam instead of Ham

Usefull instead of useful

“Water rich calcium” to be replaced by Calcium rich water

and many others

Author Response

Reviewer #2

I read with interest this manuscript aiming at determining the applicability, usability and accuracy of Nutricate, an online application assessing dietary intake in cow’s milk allergic children. The article deals with an important topic given the young population under study and the effects that elimination diets can have on it. 

I suggest modifying the title simply by deleting the term “online” form one of its appearances to avoid repetitions.

Many thanks for this commentary, the title has been corrected.

The manuscript is somehow understandable although not very well written, especially in the discussion.

The English language has been reviewed by a native English speaker. Please note other modifications highlighted in yellow have been made along the manuscript to facilitate the review.

Length of the introduction is good respect to the length of the manuscript. It reviews the background and states the objectives adequately.

Many thanks for this commentary

In the discussion the results are often described as consistent and similar to those of other studies. As a clinician, I would have liked the data obtained on the intake of the various micro and macronutrients to be commented more and not only by comparing them with those obtained with other tools.

A specific paragraph has been added to address this commentary

The conclusions respond to the aim of the study.

The references are fairly recent apart from 4 papers probably kept for their unicity.

Specific remarks

Line 9 of the introduction, please add soy formulas as other possible substitutes.

Many thanks for this commentary, “soy formulas” has been added

Line 67 I wasn’t able to get the meaning of the sentence: “All CMA diagnosis were realized at least for 4 months.”

This sentence has been rephrased.

I encountered several syntax and typo errors, such as The words micro/macronutriments should be substituted by micro/macronutrients.“were closed to …” to be replaced by “were close to …”In table 1 allergia instead of allergies Supp Table 1 Jam instead of Ham Usefull instead of useful “Water rich calcium” to be replaced by Calcium rich water and many others

The English language has been reviewed by a native English speaker.